# Sex- and Age-Related Dynamic Changes of the Macroelements Content in the Femoral Bone with Hip Osteoarthritis

**DOI:** 10.3390/biology11030344

**Published:** 2022-02-22

**Authors:** Mikołaj Dąbrowski, Anetta Zioła-Frankowska, Marcin Frankowski, Przemysław Daroszewski, Agnieszka Szymankiewicz-Szukała, Łukasz Kubaszewski

**Affiliations:** 1Adult Spine Orthopaedics Department, Poznan University of Medical Sciences, 61-545 Poznan, Poland; pismiennictwo1@gmail.com; 2Department of Analytical Chemistry, Faculty of Chemistry, Adam Mickiewicz University in Poznan, 61-614 Poznan, Poland; anettazf@amu.edu.pl; 3Department of Analytical and Environmental Chemistry, Faculty of Chemistry, Adam Mickiewicz University in Poznan, 61-614 Poznan, Poland; marcin.frankowski@amu.edu.pl; 4Department of Organization and Management in Healthcare, Poznan University of Medical Sciences, 61-545 Poznan, Poland; dyrektor@orsk.pl; 5Department Pathophysiology of Locomotor Organs, Poznan University of Medical Sciences, 61-545 Poznan, Poland; aszukala@orsk.pl

**Keywords:** structural elements, macroelements, femoral bone, Ca/P radio, ICP-OES, arthroplasty, osteoarthrosis, osteoporosis, bone mineral density (BMD)

## Abstract

**Simple Summary:**

The study assessed the content of macroelements Ca, Mg, P, and Na in the proximal femoral bone tissue in patients with hip osteoarthritis, and it correlated with age, sex, and BMI. The high reduction of macroelements in the femoral bone of patients with hip osteoarthritis is more pronounced in the cortical bone and occurs in women under 60 years of age. In men, it begins in the seventh and increases in the eighth decade of life.

**Abstract:**

Background: The content of macroelements in bones varies with age and depends on sex. The aim of the study was to evaluate the content of macroelements and its correlation with age and sex in the femoral bone obtained during total hip arthroplasty. Methods: In the 86 patients, the content of macroelements (Ca, P, Mg, and Na) in the femoral head and neck (cancellous and cortical bone) was assessed by means of the inductively coupled plasma optical emission spectrometry analytical technique (ICP-OES). Results: There was a decrease in the content of macroelements in the cortical bone with age in the women in the 51–60 years (statistically significant: −0.59 for Ca, −0.65 for P) and over 70 years age groups (correlation not statistically significant: −0.29 for Ca, −0.38 for P). A significant decrease in the content of macroelements in the cortical bone was found in men over 70 years of age. Conclusions: Patterns of increased loss of macronutrients (Ca, P, and Mg) in the femoral neck (cortical bone) were demonstrated in the following patients with osteoarthritis: women aged 51–60 years and patients of both sexes over 70 years of age.

## 1. Introduction

Bone remodeling is age- and sex-dependent—women tend to lose more bone mass with age than men [1]. In women, it begins in the third or fourth decade of life, with its maximum in the menopausal period [2]. Starting in the seventh decade of life in women and about ten years later in men, a decrease in bone mineral density (BMD) was observed with a disturbance of the bone microarchitecture, increasing the frequency of fractures [3]. At the beginning of the menopause, bone mass is lost mainly in the cortical bone, and its porosity increases; a decrease in the number of trabeculae in the cancellous bone and a general decrease in bone strength are observed as well [4]. Age-related structural changes in bone and bone loss tend to preserve mechanically loaded bone regions, while less mechanically stressed bone fragments weaken with age [5]. Structural defects in the proximal femur associated with fractures have been shown to overlap with structural changes associated with normal aging [6]. 

Patients with osteoarthritis (OA) showed a decrease in BMD and concentration of Ca and Mg [7]. A previous study showed a change in the content of structural elements with age in the head and neck of the femur [8].

The assessment of the content of macroelements (Ca, P, and Mg) in the cortical and trabecular bones depends on sex and age and may be important for the further remodeling of the bone around the implant and thus for durability after arthroplasty. Previous studies of the content of structural macroelements focused mainly on determining their change with age in women and men without determining the dynamics of changes in specific decades of life. 

The aim of the study was to assess the dynamics of changes in the content of macroelements (Ca, P Mg, and Na) in the cancellous and cortical femoral bone in the successive decades of life in women and men who had undergone primary total hip arthroplasty (THA) without fractures.

## 2. Materials and Methods

The study included 86 patients (Caucasian race, 50 women and 36 men) who underwent hip replacement at the W. Dega University Hospital, Poznan University of Medical Sciences, Department of Spine Surgery, Oncological Orthopedics, and Traumatology and Department of Orthopedics and Traumatology. The indications for this procedure in all the cases were primary degenerative changes of the hip joint in the people aged > 50 years.

All the patients included in the study showed mild, moderate, and severe stages (Kellgren–Lawrence radiographic grades 2–4) of primary OA in the hip joint. The exclusion criteria were secondary degenerative changes and hip fractures, history of cancer, liver or kidney failure, environmental exposure to toxic factors, overt endocrine disease, or previous/current therapy with anti-osteoporotic drugs or glucocorticoids.

The information was collected from each patient, which was used to fill in the questionnaire used for later statistical studies: patient’s age, sex, body mass index—BMI (one day before surgery). The study group was divided according to sex into three age groups: 51–60, 61–70, >70 years of age (Table 1). For each group, the correlation with age was analyzed separately (Spearman’s rank correlation).

### 2.1. Chemical Analysis

The biological sample preparation method applied was taken from Zioła-Frankowska et al. (2015) [9]. Anyway, it should be stressed here that before the chemical sample preparation step, the bone samples were divided into the femoral head samples and the femoral neck samples with different thickness under sterile states during the collection step. Acid digestion using microwave energy was used as the most effective method for sample preparation. After mineralization, several metals were determined using the ICP-OES analytical technique.

### 2.2. Statistical and Chemometrics Analysis

The statistical analysis was performed using Statistica 13.5 (StatSoft). The normal distribution was determined using the Shapiro–Wilk test (*p* < 0.05). The concentration of macronutrients in the bone tissue was compared between the age groups and both sexes using the Kruskal–Wallis test and the Mann–Whitney test (*p* < 0.05). The analysis of the correlation (Spearman’s rank) between the concentration of micronutrients in the femur (femoral neck and head) and age was performed in three different age groups and groups of both sexes.

## 3. Results

The Ca concentration in the women showed a similar trend of changes in the femoral head (cancellous bone) and the femoral neck (cortical bone), assuming lower values in the 51–60 years group (median, 112.1 g/kg and 148 g/kg, respectively). In the 61–70 years group, it increased to the highest values, 127.7 g/kg and 165.8 g/kg, respectively, and in the oldest age group, it decreased to the middle values of 122.4 g/kg and 139.7 g/kg, respectively (Figure 1, Table 2).

Slightly different concentrations were found in the men in the femoral head, in whom the median Ca increased from 123.8 g/kg in the 51–60 years group to 141.8 g/kg in the seventh decade of life and reached the highest value in the oldest patients—150 g/kg. On the other hand, in the femoral neck of men, the median Ca concentration significantly decreased with age, amounting to 171.4, 144.4, and 118.5 g/kg, respectively, in the successive age groups (Figure 1, Table 1).

Analogous changes in concentration for age and sex were found in the content of P. In the women, the highest concentration was found in the head and neck of the femur in the age group of 61–70 years—the median concentration was 57.4 and 73.5 g/kg, respectively. In the case of men, in the femoral neck, the P concentration decreased with age from 73.1 g/kg in the 51–60 years group to 55.2 g/kg at >70 years of age (Figure 2, Table 1).

The Ca/P ratio was similar and the differences were statistically insignificant between the sexes, the age groups, and the types of bone tissue (Table 1).

In the women, in the head and neck of the femur, the highest Mg concentrations were found in the age group of 61–70 years—the median concentration was 1.7 and 1.4 g/kg. In the men, the concentration decreased with age in the femoral neck from 1.7 to 1/3 g/kg, and in the femoral head, it increased with age from 1.3 to 1.6 g/kg (Table 1).

In the patients over 50 years of age, the evaluation of the correlation of changes in the concentration of building elements in the bone showed a significant negative correlation for Ca (−0.48) and P (−0.41) with age only in men in the femoral neck (cortical bone) (Figure 2, Table 3).

Analyzing the correlations of the content of elements in the bone tissue with age in the individual age groups, a significant decrease in the content of elements in the cortical bone with age was shown in the women in the 51–60 years group. Spearman’s coefficient was as follows: Ca, −0.59; P, −0.65; Mg, −0.63; and Na, −0.69. In the 61–70 years group, the coefficient was close to zero, and in the over 70 years age group, it was negative again, but not statistically significantly, and ranged from −0.44 for Ca to −0.22 for Mg. In men, a significant decrease in the content of Ca and P in the cortical bone with age in the age groups of 61–70 years (Ca, −0.56; P, −0.53) and over 70 years was shown (Ca, −0.71; P, −0.76) (Figure 3, Table 3).

The mean values of the BMI in the individual age groups in both women and men did not differ statistically significantly. There was an upward trend with each decade of life for women and a decreasing trend for men (Table 1).

The women showed an increase in the BMI in the successive decades of life. In the men, the BMI remained stable from 51 to 70 years of age, and in the patients over 70 years of age, it decreased (Figure 4). Analyzing the correlations of the content of elements in the bone tissue with the BMI in the individual age groups, a significant decrease in the content of elements in the cortical bone with age was shown in the women in the 61–70 years group. Spearman’s coefficient ranged from 0.53 to 0.59 (Figure 5). Removal of extreme values did not change the demonstrated correlation trends. 

## 4. Discussion

We confirmed that the content of calcium, phosphorus, and magnesium in the femoral bone changed with age and depended on sex and the type of bone tissue. We observed a reduction in the content of macroelements in the cortical bone (femoral neck) of the patients with advanced degenerative changes in both women and men.

Brodziak et al. showed the content of calcium and phosphorus in the cortical bone of the femur decreased with age, but also in the intertrochanteric cancellous bone. In all the parts of the hip joint, the calcium content differed significantly between the age groups [10]. Karaaslan et al. observed a tendency for the concentration of Ca and P to decrease with age in women and men [7]. Jurkiewicz et al. also reported a decrease in the calcium content with age [11]. One study showed that the levels of calcium and phosphorus in older women did not change with age [12].

Karaaslan et al. showed a significantly negative correlation between BMD and age [7]. It was also confirmed that mechanical properties and microstructural parameters are correlated with BMD [13]. Berger et al. showed that BMD loss in the femur begins at medium of age 45 and is maximal at age 50–54. This accelerated loss is a major determinant of differences in bone loss patterns between men and women. Then, at the age of 55–70, there is a period of reduction in BMD loss, and then after the age of 70, bone loss is accelerated, reaching its maximum acceleration at the age of 70–85 years. The latter period of rapid bone loss in women as well as in men may play a role in the increased incidence of hip fractures among older patients [14,15]. In our study, we additionally assessed the previously unknown dynamics of changes in the Ca and P content in the femur, which differed between age groups and between men and women. Women in the 51–60 years age group had a reduced content of structural elements compared to older patients, and their correlation with age was significantly negative in this group. Our study of changes in the content of femoral macronutrients (Ca, P, and Mg) documents patterns in the rate of increase and loss of femoral BMD and differences between men and women.

In patients aged ≥70 years qualified for THA, a high frequency of decreased BMD was found [16]. Our study confirms the reduced mineral content in this age group; at the same time, a bone density assessment should be considered before performing THA. On the other hand, studies indicate no correlation between the BMD of the femoral neck and cementless loosening after THA in 2-year follow-up in patients under 75 years of age without osteoporosis [16]. Patients with OA have been shown to have higher BMD compared to control, which is probably associated with increased sclerotization and remodeling within the joint [17]. It seems that severe hip OA may be associated with higher BMD, but only in the femoral neck. A possible explanation for this process is that the greater trochanter and the femoral neck show different BMD responses depending on the degenerative load condition of the degenerated hip joint. In severe stages of hip osteoarthritis, decreased mobility of the joint can lead to a reduction in trochanteric BMD by impairing muscle function and muscle attachments. This may in turn overload the femoral neck and increase BMD by reducing the transmission of muscle forces [18,19]. This can be confirmed by studies which found that OA does not protect against fractures of the proximal femur, but is associated with a change in the location of the proximal femur fracture, increasing the likelihood of trochanteric fractures [20]. The explanation for the different correlations in the age groups among women is the BMI showed a significant correlation with macroelements in the group of women aged 61–70, the same group where there was a significant positive correlation between building elements and age. This may indicate that body weight may have been an additional factor in this group.

In the men, the BMI was at a similar level from 51 to 70 years of age and clearly decreased in the patients aged over 70 years. The studies conducted so far have shown a possible relationship of the BMI with the risk of OA of hip joints [21]. There were differences in the BMI in specific decades of life between the men and the women. This observation may be explained by the association of obesity with increased pain intensity and worse functioning in women waiting for hip arthroplasty despite the radiological severity of the disease being similar to that in men [12].

Some studies emphasize that the introduced diet among obese postmenopausal women with osteoarthritis awaiting arthroplasty resulted in significant weight loss and an increase in circulating vitamin D. Among the patients who lost weight, an increase in bone resorption parameters was noted despite a more marked increase in plasma vitamin D and a greater increase in dietary protein content [22]. One of the factors influencing the Ca and P content may be the diet. It was shown, however, that increasing dietary calcium intake and calcium supplements caused a slight increase in BMD [23]. Whether it is due to estrogen deficiency, decreased physical activity, or something else entirely remains to be explored [24]. 

There is no clear relationship between postmenopausal hormonal disorders, the age of the menopause, and the use of the estrogen replacement therapy and hip arthroplasty in hip osteoarthritis [25,26,27].

There was no statistically significant change in the Ca/P ratio in the trabecular bone of the femur with age, nor were there any significant sex differences [28,29]. Another study confirmed that the Ca/P ratio did not change with age [30]. We found similar results in this study, in which the Ca/P ratio of 2.2 (range, 2.1–2.3) was constant regardless of sex and age.

The content of magnesium in the femur also changes with age and is dependent on sex and the type of bone tissue, similarly to Ca and P. We observed a reduction in the Mg content in the cortical bone (femoral neck) of both female and male patients. Other authors also observed a decrease in the Mg content in the femoral bone with the age of patients with OA [7,11,28,29]. The Mg content in the bone tissue was significantly correlated with BMD and was significantly lower in the group with fractures in women [7]. Mg plays a role in the metabolism of minerals in the bone tissue, mimicking the action of Ca in transport and mineralization, and is also involved in the absorption of Ca.

The clinical implications of our study indicate the possibility of coexistence of osteoporotic changes, and the measure may be the content of macronutrients in the neck of the femur. We did not show significant correlations of macronutrients and their changes with age in the femoral head (cancellous bone), which indicates its lower influence on bone metabolism of OA.

The limitation of the study is a fairly small number of patients. Moreover, due to the limited availability of healthy bones (the femur is not removed in healthy people, and the bone material from cadavers is difficult to access), we did not have a control group. Therefore, we compared the results with the data of other publications. Moreover, the content of elements depends on enzymatic, oxidative, and environmental factors that are difficult to exclude. Therefore, it was not possible to exclude all sources of the elements despite the fact that the bone material belonged to patients coming from a fairly homogeneous environment. Finally, the conclusions drawn from the conducted study related to metabolic changes in osteoarthritis are indirect.

This study emphasizes the need for a controlled study of the causes of bone degradation in patients with OA, including its dynamics in relation to bone density, and comparison to osteoporosis and healthy groups and the need to assess physical activity and bone turnover markers. 

## 5. Conclusions

The more intensive reduction of macroelements (Ca, P, and Mg) in patients with OA measured quantitatively by means of inductively coupled plasma emission spectrometry is more pronounced in the cortical bone and occurs in women before the age of 60. Body weight and the interaction of OA and osteoporosis on bone metabolism are potential factors of age-dependent variability. In men, this reduction begins in seventh and increases in the eighth decade of life.

## Figures and Tables

**Figure 1 biology-11-00344-f001:**
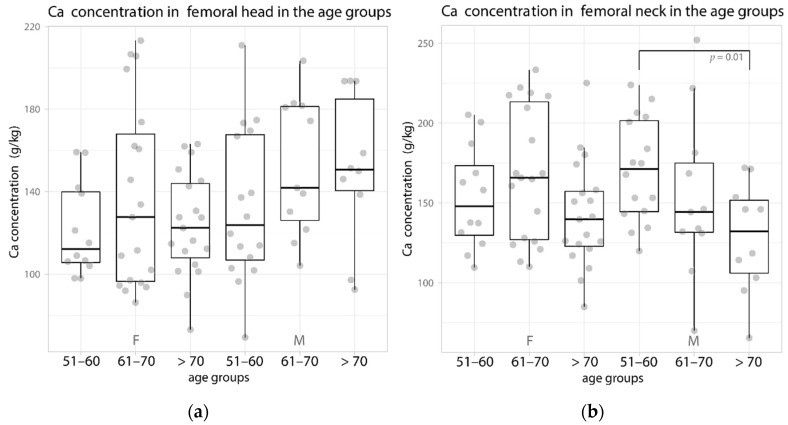
Data distribution (jittered dots) and boxplot of the median Ca content (in g/kg) in the age groups in women and men in the femoral (**a**) head and (**b**) neck. F—female, M—male.

**Figure 2 biology-11-00344-f002:**
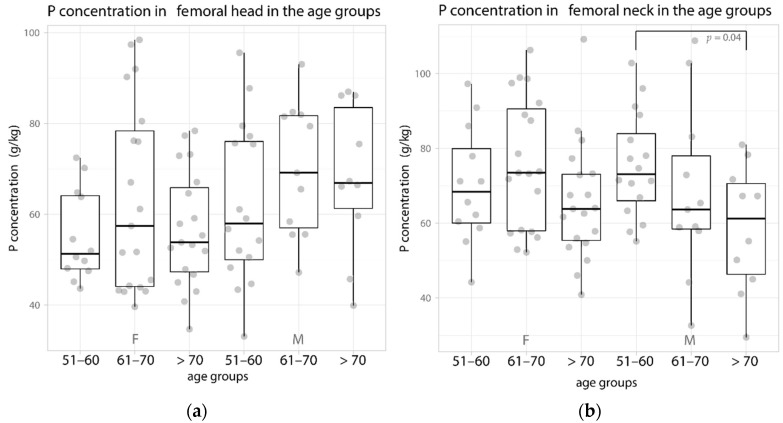
Data distribution (jittered dots) and boxplot of the median P content (in g/kg) in the age groups in women and men in the femoral (**a**) head and (**b**) neck. F—female, M—male.

**Figure 3 biology-11-00344-f003:**
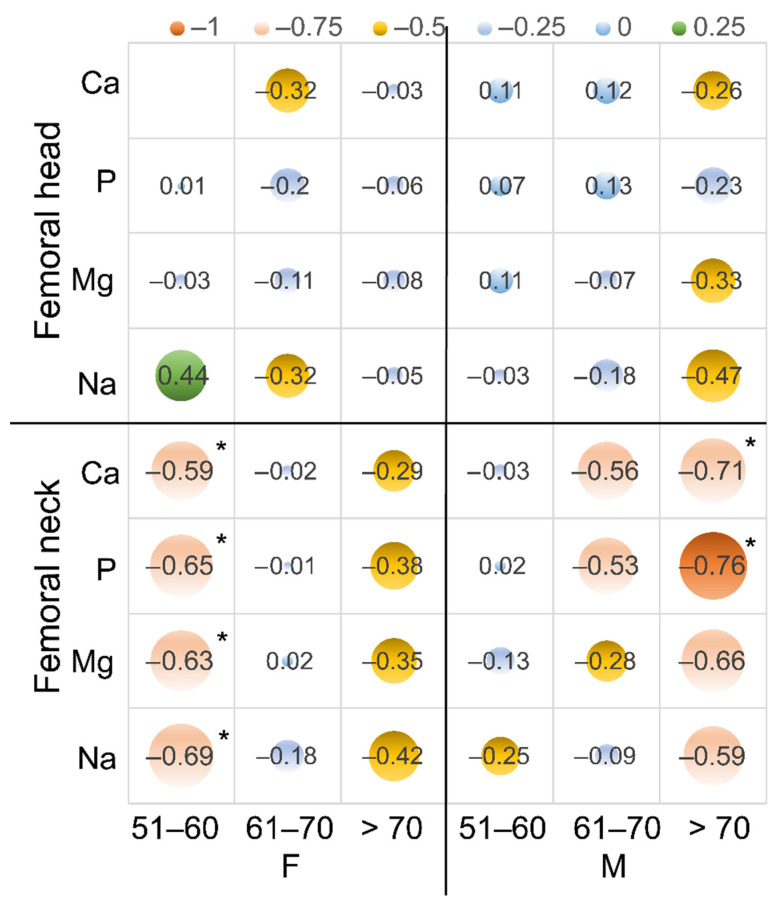
The bubble chart is a variation of Spearman’s correlation coefficient for the content of Ca, P, Mg, and Na with age in the femoral head and neck and age in the age groups. An additional dimension of the data is represented in the size of the bubbles; *: statistically significant.

**Figure 4 biology-11-00344-f004:**
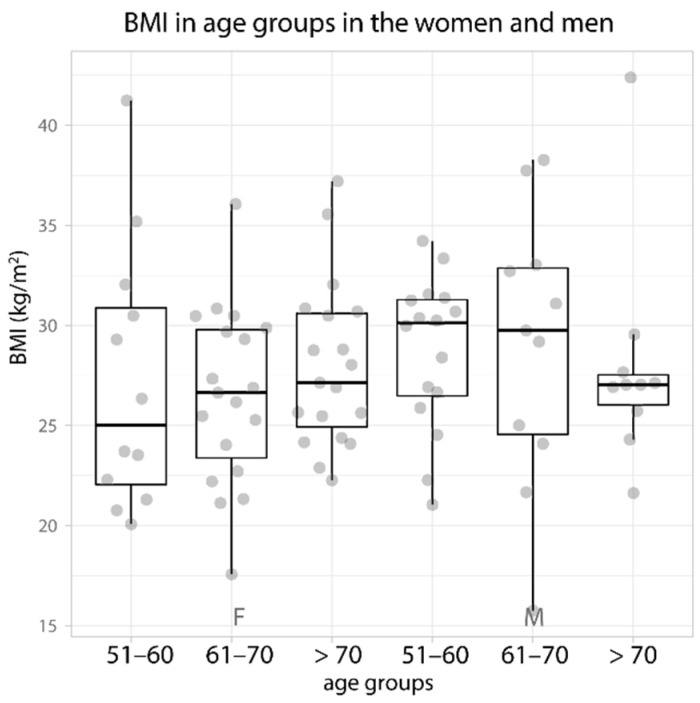
Data distribution (jittered dots) and boxplot of the median BMI (kg/m^2^) in the age groups in the women and men. The data are ordered according to their median value. F—female, M—male.

**Figure 5 biology-11-00344-f005:**
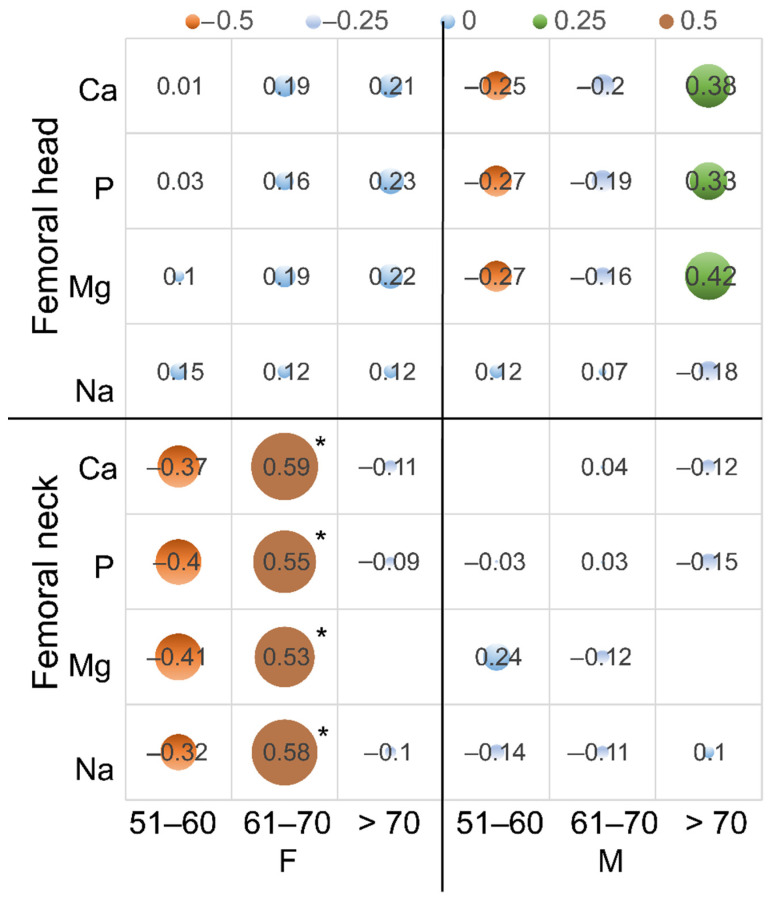
The bubble chart is a variation of Spearman’s correlation coefficient for the content of Ca, P, Mg, and Na with the BMI in the femoral head and neck and age in the age groups. An additional dimension of the data is represented in the size of the bubbles; *: statistically significant.

**Table 1 biology-11-00344-t001:** Concentrations of elements (in g/kg on the dry mass basis) (mean ± SD and median (Q1–Q3)) in the femoral head and neck in the sex and age groups (*N* = 86).

Sex		Female			Male		
Age Groups		51–60	61–70	>70	51–60	61–70	>70
*n*		12	19	19	16	11	9
Femoral neck	Ca	153.4 ± 32.3 148 (128–178)	168.6 ± 41.7 165.8 (126–217)	142.3 ± 33.2 139.7 (121–158)	170.8 ± 32.4 171.4 (144–202)	153.6 ± 50.9 144.4 (131–181)	126.7 ± 36.5 118.5 (103–154)
	P	70.1 ± 15.6 68.4 (60–82)	76 ± 17.6 73.5 (58–92)	65.6 ± 15.8 63.8 (55–73)	75.5 ± 13.9 73.1 (65–86)	68.1 ± 22.9 63.7 (58–83)	57.7 ± 17.9 55.2 (45–72)
	Ca/P	2.2 ± 0.1 2.2 (2.1–2.2)	2.2 ± 0.1 2.2 (2.1–2.3)	2.2 ± 0.1 2.2 (2.1–2.2)	2.3 ± 0.2 2.2 (2.2–2.3)	2.3 ± 0.1 2.3 (2.2–2.3)	2.2 ± 0.1 2.2 (2.1–2.3)
	Mg	1.6 ± 0.3 1.6 (1–2)	1.7 ± 0.3 1.7 (1–2)	1.5 ± 0.3 1.5 (1–2)	1.7 ± 0.3 1.7 (1–2)	1.5 ± 0.4 1.6 (1–2)	1.4 ± 0.3 1.3 (1–2)
	Na	4.6 ± 0.9 4.5 (4–5)	4.9 ± 0.8 5 (4–5)	4.6 ± 0.7 4.7 (4–5)	4.8 ± 1 4.6 (4–5)	4.5 ± 1.1 4.6 (4–6)	4.3 ± 1.1 3.7 (4–5)
Femoral head	Ca	121.5 ± 22.6 112.1 (105–141)	137.4 ± 44.8 127.7 (96–174)	124 ± 25.2 122.4 (105–145)	132.9 ± 37.2 123.8 (106–168)	152.3 ± 33.3 141.8 (122–182)	146.9 ± 35.4 150 (139–159)
	P	55.2 ± 10 51.3 (48–64)	63.3 ± 20.7 57.4 (44–81)	56.6 ± 12.7 53.8 (47–67)	62.1 ± 17.7 57.9 (49–76)	70 ± 14.6 69.1 (56–82)	66 ± 16.1 66.5 (60–75)
	Ca/P	2.2 ± 0.1 2.2 (2.2–2.2)	2.2 ± 0.1 2.2 (2.1–2.2)	2.2 ± 0.1 2.2 (2.1–2.2)	2.2 ± 0.3 2.2 (2.2–2.2)	2.2 ± 0.1 2.2 (2.1–2.2)	2.2 ± 0.1 2.2 (2.2–2.3)
	Mg	1.3 ± 0.3 1.2 (1–1)	1.5 ± 0.4 1.4 (1–2)	1.3 ± 0.3 1.3 (1–2)	1.4 ± 0.3 1.3 (1–2)	1.6 ± 0.3 1.6 (1–2)	1.5 ± 0.4 1.6 (1–2)
	Na	5.6 ± 1 5.4 (5–6)	5.9 ± 1.1 5.8 (5–6)	4.9 ± 0.9 4.8 (4–5)	5.2 ± 0.9 5.2 (4–6)	5.7 ± 1.3 5.8 (5–7)	5.7 ± 1.2 5.8 (6–7)

**Table 2 biology-11-00344-t002:** Number of patients, BMI (kg/m^2^), and weight (kg) (mean ± SD and median (Q1–Q3)) of the patients in the sex and age groups (*N* = 86).

Sex	Female			Male		
Age Groups	51–60	61–70	> 70	51–60	61–70	>70
*n*	12	19	19	16	11	9
Weight (kg)	71.1 ± 18.1 69 (56.3–81)	69 ± 14.5 70 (56–77)	69.9 ± 10.7 69 (64.4–73)	87 ± 12.8 89 (77–95)	86.1 ± 22.1 92.5 (68–106)	78.6 ± 15.3 77 (71–80)
BMI (kg/m^2^)	27.2 ± 6.6 25 (21.8–31.3)	26.5 ± 4.4 26.6 (22.7–29.9)	28 ± 4.1 27.1 (24.4–30.7)	28.7 ± 3.8 30.1 (26.3–31.3)	28.9 ± 6.8 29.8 (24.1–33)	28 ± 5.8 27 (25.7–27.7)

**Table 3 biology-11-00344-t003:** Spearman’s correlation coefficient for the content of Ca, P, Mg, and Na in the femoral head and femoral neck and age in the age groups (*N* = 86).

Sex		Female				Male			
Age Groups		51–60	61–70	>70	All	51–60	61–70	>70	All
*n*		12	19	19	50	16	11	9	36
Femoral neck	Ca	−0.59 *	−0.02	−0.29	−0.21	−0.03	−0.56	−0.71 *	−0.48 *
	P	−0.65 *	−0.01	−0.38	−0.21	0.02	−0.53	−0.76 *	−0.41 *
	Mg	−0.63 *	0.02	−0.35	−0.20	−0.13	−0.28	−0.66	−0.44 *
	Na	−0.69 *	−0.18	−0.42	−0.14	−0.25	−0.09	−0.59	−0.30
Femoral head	Ca	0	−0.32	−0.03	−0.08	0.11	0.12	−0.26	0.23
	P	0.01	−0.2	−0.06	−0.08	0.07	0.13	−0.23	0.15
	Mg	−0.03	−0.11	−0.08	−0.05	0.11	−0.07	−0.33	0.15
	Na	0.44	−0.32	−0.05	−0.35 *	−0.03	−0.18	−0.47	0.12

* Statistically significant.

## Data Availability

The data presented in this study are available on request from the corresponding author.

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
