# Peer review of "Sex- and Age-Related Dynamic Changes of the Macroelements Content in the Femoral Bone with Hip Osteoarthritis"

_biology, 2022, doi:10.3390/biology11030344_

Round 1

Reviewer 1 Report

The manuscript has been significantly improved compared to the initial submission. The significance of work was more pronounced and the authors did a great job in explaining the results in the discussion session. Several minor things can be improved.

1) The authors use a couple of statistical analyses in the manuscript to support their conclusion. In the method part, could you explain in more details on those methods and formulas involved in getting the numbers in the manuscript?

2) In Figure 1a, is the Y-axis legend 'Ca concentration'? A typo here could be misleading.

Author Response

Dear Revier 1,

The manuscript has been significantly improved compared to the initial submission. The significance of work was more pronounced and the authors did a great job in explaining the results in the discussion session. Several minor things can be improved.
We thank you for taking your time and effort to provide helpful and constructive comments. We have provided point-by-point answers to the specific comments raised by the reviewers and the modifications have been applied to the new “revised manuscript”. 

1) The authors use a couple of statistical analyses in the manuscript to support their conclusion. In the method part, could you explain in more details on those methods and formulas involved in getting the numbers in the manuscript?
The method has been corrected.

2) In Figure 1a, is the Y-axis legend 'Ca concentration'? A typo here could be misleading.
The figure has been corrected.

Reviewer 2 Report

Overall comment:

The authors addressed many issues by the reviewers. The current manuscript still has many grammar issues and is missing parts in the discussion section. At the current stage, it still needs revisions before publication.

  1. The explanation on the temporal change comparisons in the 3rd paragraph of discussion “the lack of correlation with age may indicate a more advanced stage of the degenerative disease” is less convincing, since the maroelement concentration in 61-70 and >70 groups show small decline or even increase in some groups. Discussion regarding the decade changes should refer to previous literature on BMD or BMC temporal changes with age such as PMID: 18559803, 8931029

Specific comments:

  1. Rearrange the words order in the title from “contents macroelements” to “macroelements contents”
  2. Check grammar of the sentence in the summary, “The study was assessment the content…”. The grammar should be “The study was assessing…” or “The study assessed…”
  3. Abstract, the correct grammar should be “Assessment of the content of elements Ca… was performed by …”
  4. Page 2, materials and methods section. Should be “mild, moderate and severe stages”
  5. Page 3, try to cite table 1 before table 2. Otherwise reorder the table number. The data format is not clearly explained. Please add the same statement as table 2 “mean±SD and median (Q1-Q3)”.
  6. Table 2, typo in “neckin”
  7. Page 3, section 2.2, should be “in 3 different age groups and both sex groups”
  8. Page 4, figure 1 (a), y axis label should be “Ca concentration” rather than “P concentration”
  9. Figure 5, check the data format. Should be “.” not “,”
  10. The logic in the discussion session should be clearer. The authors may think about rearranging/organizing certain paragraphs to show a clear logic flow.

Author Response

Dear Reviewer 

We thank you for taking your time and effort to provide helpful and constructive comments. We have provided point-by-point answers to the specific comments raised by the reviewers and the modifications have been applied to the new “revised manuscript”.

The explanation on the temporal change comparisons in the 3rd paragraph of discussion “the lack of correlation with age may indicate a more advanced stage of the degenerative disease” is less convincing, since the maroelement concentration in 61-70 and >70 groups show small decline or even increase in some groups. 
The discussion has been corrected.

Discussion regarding the decade changes should refer to previous literature on BMD or BMC temporal changes with age such as PMID: 18559803, 8931029
The discussion has been corrected with considering PMID 18559803, 8931029  

Specific comments:
1.    Rearrange the words order in the title from “contents macroelements” to “macroelements contents”
The title has been corrected.

2.    Check grammar of the sentence in the summary, “The study was assessment the content…”. The grammar should be “The study was assessing…” or “The study assessed…”
The text has been corrected.

3.    Abstract, the correct grammar should be “Assessment of the content of elements Ca… was performed by …”
The text has been corrected.

4.    Page 2, materials and methods section. Should be “mild, moderate and severe stages”
The text has been corrected.

5.    Page 3, try to cite table 1 before table 2. Otherwise reorder the table number. The data format is not clearly explained. Please add the same statement as table 2 “mean±SD and median (Q1-Q3)”.
The text has been corrected.

6.    Table 2, typo in “neckin”
The text has been corrected.

7.    Page 3, section 2.2, should be “in 3 different age groups and both sex groups”
The text has been corrected.

8.    Page 4, figure 1 (a), y axis label should be “Ca concentration” rather than “P concentration”
The figure has been corrected.

9.    Figure 5, check the data format. Should be “.” not “,”
The figure has been corrected.

10.    The logic in the discussion session should be clearer. The authors may think about rearranging/organizing certain paragraphs to show a clear logic flow.
The discussion has been corrected.

Round 2

Reviewer 2 Report

Overall comment:

The authors addressed the issues raised by the reviewers. The authors can go through the manuscript with a final grammar check before publication.

Specific comments:

  1. Check grammar of the sentence in the summary. The grammar should be “The study was assessing…” or “The study assessed…”
  2. Abstract, the correct grammar should be “Assessment of the content of elements Ca… was performed by …”
  3. Page 6, why is table 2 arranged after table 3? Otherwise reorder the table number.
  4. The authors may think about rearranging/shortening certain paragraphs to show a clear logic flow in the discussion session.

This manuscript is a resubmission of an earlier submission. The following is a list of the peer review reports and author responses from that submission.

Round 1

Reviewer 1 Report

In the manuscript by Dabrowski et al., the authors did analyses on the changes of different minerals in the femoral bone of men and women with hip osteoarthritis. The manuscript clearly present the contents of such elements among 86 patients. The data looks solid, but the fundamental design and analyses may require further examination.

  1. there is no information regarding the heterogeneity of the patients under analysis. Information such as races, weight, BMI, severity of the osteoarthritis etc are completely unknown.
  2. from figure 4, at least we know the BMI is quite heterogenous within age groups. How would that affect the structural elements' contents overall? Could you stratify the patients and redo the analysis to see if the same trends hold?
  3. in age group 51-60, there was a negative correlation between the structural elements and the age, could you provide a possible explanation for it?
  4. what do these results mean clinically?

Reviewer 2 Report

Summary:

In this manuscript, the authors checked four element contents in patients’ proximal femur and their correlation with each age decade and BMI. This manuscript is particularly weak in several aspects. The selection of age groups and how are they correlated to menopause are missing from the discussion section. Discussions regarding the difference of observed element content and correlation coefficients between femur neck and femur head are completely missing.

Comments:

This manuscript has many grammar issues and is missing critical information in the method and discussion sections. The observed content change and correlation within each decade in women and men are inadequately discussed regarding regional differences, selection of the 4 elements, the element-wise difference and the probable cause, and clinical relevance. At the current stage, it is not suited for publication.

  1. Since all the patients are with severe OA, the OA condition should not only be mentioned in the drawbacks of the manuscript but should be stressed in the discussion. How did the observed element content compare with normal health cases in literature? Without a control group or control region, the observed changes can not be ruled purely due to age effects.
  2. Confusing term of “structural element”, which could lead to misunderstanding of trabecular structural differences. Use element instead.
  3. It’s very confusing how the correlation coefficient is calculated. Please clarify that in the method section. For instance, is age using the actual age number or subtracted with the beginning year of each decade? How would using the actual age rather than absolute age changes within each decade affect the outcome?
  4. The sample preparation steps are completely missing in the methods section. Are samples sectioned, polished and how are the ROI and orientations selected from the samples?

Specific comments:

  1. Title, too many ‘of’s in the title. Maybe change “dynamics of changes” to “dynamic changes”
  2. Check grammar of the sentence in the summary, “The study was determined the content…”
  3. Check grammar of the sentence in the abstract, “Determine the content of elements Ca… was performed by …”
  4. Typo in “I In cancellous bones in men…” in abstract
  5. Should be coefficient not coefficients since only group over 70 years of age is mentioned in the sentence. “… negative but not significant correlation coefficients were shown in the group over 70 years of age”
  6. Keep the terminology (cortical bone” and “compact bone”, ”spongy bone” and “cancellous bone”) consistent through the context. “
  7. Page 2, check grammar in the sentence. “In patients with osteoarthritis showed a decrease in BMD and calcium …”
  8. Page 2, before using the term OA, the full name should be indicated first in the context.
  9. Page 5, table 2, please check the table lines and make it complete.
  10. The IRB statement and informed consent statement should also be included in the methods section.
  11. The decade term, especially the “next decade” is very confusing in the context. Please change it to numerical values such as 20-30s.
  12. Typo “and” in the discussion
  13. Demographic information missing (race and gender distribution) in the method section.
  14. BMI measurement time is missing in the method section.
  15. Please indicate the significance on top of the box plot